# The Role of Dishware Size in the Perception of Portion Size in Children and Adolescents with Obesity

**DOI:** 10.3390/nu13062062

**Published:** 2021-06-16

**Authors:** Annica Franziska Dörsam, Alisa Weiland, Helene Sauer, Katrin Elisabeth Giel, Nanette Stroebele-Benschop, Stephan Zipfel, Paul Enck, Isabelle Mack

**Affiliations:** 1Department of Psychosomatic Medicine and Psychotherapy, University Hospital Tübingen, 72076 Tübingen, Germany; Annica.Doersam@med.uni-tuebingen.de (A.F.D.); alisa.weiland@med.uni-tuebingen.de (A.W.); helene.sauer@med.uni-tuebingen.de (H.S.); katrin.giel@med.uni-tuebingen.de (K.E.G.); stephan.zipfel@med.uni-tuebingen.de (S.Z.); paul.enck@uni-tuebingen.de (P.E.); 2Institute of Nutritional Medicine, University of Hohenheim, 70599 Stuttgart, Germany; N.Stroebele@uni-hohenheim.de

**Keywords:** children, adolescents, obesity, overweight, dishware, portion size

## Abstract

Purpose: The influence of dishware on portion size perception in children and adolescents is inconclusive. This study investigated how children and adolescents with both obesity and a normal weight perceived portion size in different sized and shaped dishware items. Methods: The study included 60 children and adolescents with overweight and obesity (OBE) and 27 children and adolescents with normal weight (NW) aged from 9 to 17 years. The participants estimated quantities in three pairs of drinking glasses, one pair of bowls and two pairs of plates which varied in size and shape. The children were instructed to state intuitively which portion they would choose for big or small thirst/hunger. Thereafter they were asked to determine the exact amount by answering which dishware item contained the larger/smaller portion (cognitive evaluation). Results: There were no substantial differences in the intuitive evaluation of portion sizes between OBE and NW. During the cognitive evaluation, OBE estimated the amount of water in the glasses more correctly compared to NW (61% vs. 43%; *p* = 0.008); OBE estimated the amount of lentils in the bowls and on the plates significantly less correctly (39%) compared to NW (56%; *p* = 0.013). Conclusions: Habit formation and environmental stimuli might play a greater role in estimating food amounts in dishware than the child’s and adolescent’s body weight.

## 1. Introduction

Childhood obesity rates have increased at an alarming rate despite intensive efforts to reduce this global epidemic [1]. Obesity is a complex and multi-factorial disease, however, the main cause of obesity in children is a positive energy balance due to caloric intake exceeding caloric expenditure, combined with a genetic predisposition to weight gain, comorbidity with mental disorders, stigma, a difficult relationship with peers, family, and other environmental factors [2,3]. The principles of weight management aim to reduce energy intake by improving diet and eating habits, i.e., modifications in the macronutrient distribution of energy intake, and to enhance energy expenditure by increasing physical activity and avoiding a sedentary lifestyle [4]. From a nutritional point of view, portion size and energy density [5,6] of foods as well as eating frequency [7] are key factors of weight management [8]. In view of the increase in portion sizes over recent decades, it is important to be familiar with appropriate food portion sizes [9]. Experimental studies have demonstrated that preschool-aged children consume more food when served larger portions [10,11,12]. A comprehensive review of mechanisms that underlie the response to portion size can be found in recently published papers [13,14,15]. In the home setting, caregiver serving sizes are strongly associated with the amounts children consume [16]. In this context, a promising and easily modifiable approach in the prevention and treatment of childhood obesity presents the immediate eating environment of children [17], more specifically, size-related features of the child’s eating environment, such as plates, cups, or bowls in which food and beverages are consumed [18]. Meta-analyses exploring the effects of modified dishware sizes on portion selection and intakes are inconsistent due to a large amount of heterogeneity across studies (e.g., size and type of dishware items, main meal vs. non-main meal/snack food), finding no effect across age groups (15 comparisons; plates and bowls) [19] and small-to-moderate effects in adults (19 comparisons; plates, bowls, cutlery, or glasses) [20]. A broader meta-analysis involving 56 comparisons across age groups concluded that the effect of altered dishware depends on how the portion size is determined, whereby smaller plates only resulted in smaller meals when the portion was self-served [21]. More precisely, the included studies in this meta-analysis which assessed children show that larger plates and bowls resulted in larger self-served portions for unit entrées, amorphous entrées and a fruit side dish, but interestingly not for a vegetable side dish in a buffet-type line at school lunch in forty-two first-grade children [18]. The children consumed nearly 50% of the calories that they served themselves [18]. Similarly, the behavioral laboratory study by Fisher et al. [22] investigated self-served entrée portions at dinner meals in sixty 4- to 6-year-old children. The results revealed that the children served themselves more of a pasta entrée when the amount available and the serving spoon size were increased; however, the larger serving size did not directly translate into larger intakes [22]. In a novel analysis of the perceptions of dishware size in households with children and adolescents, Mack et al. found that large plate and bowl sizes, but not drinking glass shapes, were associated with lower body weight in children [23,24]. To conclude, findings related to the role of dishware on children’s and adolescent’s food intake remain inconclusive. However, the type of food, self-serving vs. being served and real vs. hypothetical consumption seem to be important influencing factors [18,22,23,25,26,27]. Moreover, the children’s ability to estimate the satiating properties of foods [28] might also represent an important factor when choosing dishware as well as portion sizes.

With regard to obesity-related family environment factors, the question arises as to whether or not a relationship exists between children’s body weight and the size of the dishware used, and therefore the (self-) served portion size. The aim of this experimental study was to analyze the estimation of liquid and solid food quantities presented in/on different dishware in children and adolescents with overweight and obesity (OBE) in comparison to children and adolescents with normal weight (NW). We hypothesize that there will be differences between NW and OBE in the estimation of liquid and solid food quantities in drinking glasses, on plates and in bowls, as it is suggested that there are different obesity-related family environment factors. For example, food availability and food parenting practices (e.g., rules, monitoring, structure, modeling practices) [29,30] might represent family environment factors that contribute to children with OBE being less able to estimate food quantities. Thus, it was tested whether or not OBE and NW differentiate in their estimation of liquid/solid quantities in/on differently shaped drinking glasses, bowls and plates based on their intuitive and cognitive evaluation of the amounts.

## 2. Methods

### 2.1. Participants and Study Information

The study protocol was approved by the Ethics Committee of the medical faculty for the University Tübingen, Germany (#444/2011BO1) and is registered in the German Clinical Trials Registry (DRKS; #DRKS00005122). The present study was conducted as part of the DROMLIN-study (PreDictor Research in Obesity during Medical care—weight loss in children and adolescents during an in-patient rehabilitation) [31]. Prior to inclusion, project information sheets were disseminated to children and parents and written informed consent was obtained by the parent and assent by the child. The study took place from 2012 to 2013.

Sixty children and adolescents with overweight and obesity (OBE) with a body mass index (BMI) over the 85th percentile for their age and gender-specific norms [32], between 9 and 17 years of age (47% male) and undergoing an in-patient weight loss intervention at the Children Rehabilitation Hospital for Respiratory Diseases, Allergies and Psychosomatics, Wangen i.A., Germany, were included. The program comprised physical activity, cognitive behavioral therapy and a balanced diet. The present study was conducted during their first week of in-patient stay. Five children were considered overweight, which represents <10% of the OBE group. Therefore, only the category obesity is used in the following text. Twenty-seven healthy children with normal weight (NW) with a BMI between the 10th and ≤85 th BMI percentile and aged between 11 and 14 years (56% male) living in the area around the University Hospital Tübingen were included as a control group. Exclusion criteria were severe psychological comorbidities, linguistic or intellectual limitations, type-1 diabetes, malignant tumors, systemic disorders, or severe cardiovascular diseases. See Table 1 for further details of the study population.

### 2.2. Study Procedure

The procedure took place in a quiet, approximately 20 m^2^ large room, with each individual separately. Only the investigator and the child were in the room during the assessment. In pairs, a variety of differently sized dishware containing either the same or unequal amounts of liquids/solids were presented to OBE and NW in a standardized manner. The procedure started in the afternoon around 15:00, approximately 2 h after lunch and took 30–45 min. The participants were informed about the purpose of the study.

In detail, the dishware included three different pairs of glasses, one pair of white bowls and two pairs of white plates. The drinking glasses differed in shape and volume, and the bowls and plates differed in diameter in order to test the children’s and adolescent’s ability to estimate portions in a diversity of dishware items. The pairs contained either the same or unequal amounts of liquids (water) or solids (dried red lentils). Dried red lentils as a standardized and hygienically non-problematic quantity representative were used. See Table 2 for more details on the appearance of the drinking glasses and dishes, and the quantity ratios between the different pairs.

The experiment was separated into two steps: Firstly, differences in the intuitive evaluation (IE) of portion sizes in/on differently shaped dishware items for big or small thirst/hunger were investigated. Secondly, differences in the cognitive evaluation (CE) of the amount of portion sizes in/on differently shaped dishware items were tested. The assessment started with the glass pairs followed by the dishes. In each category, the children were allowed to choose the order by individual preference. Thus, the order was intermixed by the selection of the participants.

For assessing IE, all children and adolescents were asked the following questions one after the other for each dishware pair: “Which glass would you choose if you were very thirsty/not very thirsty?”, or respectively, “Which bowl/plate would you choose if you were very hungry/not very hungry?”. They were then asked an open question for the reason why this particular drinking glass, plate or bowl was chosen and the answer was recorded and subsequently grouped into one of the following four categories: no specific reason, easy handling/habit, volume/amount smaller, volume/amount larger.

For assessing CE, the children and adolescents were asked to evaluate for each dishware pair whether or not the amount in each of the dishware items was similar or different. Next, the CE of the portions was investigated at a quantitative level (quantitative CE, qCE): If the children answered that the amount was different within a pair, the children and adolescents were instructed to make the pair have equal amounts by transferring the extra water/dried lentils from the larger quantity container to the smaller quantity container until they determined the pair to have equal quantities (transfer task). Then, the absolute amounts of transferred extra water/dried lentils were recorded.

As a validation step, each child’s cognitive ability to distinguish between size dimensions of three everyday objects was tested (mobile phone (9 cm), book (24 cm), bottle (34 cm)). No statistical differences were observed for any of the objects and for the mean ratio for all objects between OBE and NW, respectively [23].

### 2.3. Statistics

All analyses were conducted with IBM SPSS Statistics version 26 (IBM Corp. Released 2019. IBM SPSS Statistics for Windows, Version 26.0. Armonk, NY: IBM Corp). The sample size was calculated for the clinical part of the DROMLIN-study which aimed at identifying predictors which play a role in successful weight loss and weight loss maintenance in children and adolescents [31]. However, the sample size of *n* = 87 allowed to test for medium effect sizes of Φ = 0.35 (2 × 2 χ^2^-test, α = 0.05, Power = 0.8) in this study as calculated with G-Power (3.1). Data were analyzed per protocol with <5% of missing values in the dataset due to time and logistical reasons or early treatment termination. Differences in sample characteristics between OBE and NW were calculated using unpaired *t*-tests (age, weight, height, BMI, BMI-SDS) and Chi-square test (sex).

The Chi-square test was used to calculate differences between NW and OBE in the frequency of (i) selection of dishware items for big/small thirst or hunger (IE), (ii) reasons for a specific choice (IE), and (iii) the CE of quantity.

For the analysis of CE, it was defined that for each pair of dishware, the dishware item with less content was the reference container if unequal amounts were presented. Therefore, the results of the CE were recorded as dichotomous (1 = correct, 0 = false). A CE-score across all three pairs of drinking glasses and the three pairs of dishes, respectively, was calculated by summarizing the frequency of correct answers.

To further analyze the children’s cognitive evaluation of the portions at a quantitative level (qCE), the absolute amounts of transferred water/lentils were put into context with the reference container yielding in the total difference of quantity. For example (see Figure 1): if a child said that the amount on the reference plate 2 (100 g) was less than on plate 1 (130 g), then the child was correct in its CE. During the transfer task, if the child transferred 40 g, it therefore underestimated the amount on the reference plate by 10 g (30 g difference −40 g transferred = −10 g deviation). Finally, the percentage of deviations for these absolute amounts was calculated in context with the reference container. A qCE-score across all three pairs of drinking glasses and the three pairs of dishes, respectively, was calculated by summarizing the percentage of deviation. The Mann–Whitney-U test was used to calculate differences in the qCE-score between NW and OBE. A *p*-value of <0.05 was considered significant.

## 3. Results

### 3.1. Assessment of the Amounts in Different Drinking Glasses

Pair 1. When the children had to choose the glasses according to big and small thirst (IE), significantly more NW chose the wide parabolic glass for big thirst (χ² (1, *n* = 87) = 5.98, *p* = 0.014). There was no significant difference between NW and OBE for choosing the high parabolic glass for small thirst. The majority of children intuitively decided correctly that the wide parabolic glass contained more water. When asked which glass actually contained more or less water (CE), the majority of NW and OBE answered wrong in contrast to their (correct) intuitive feeling (see Table 3). The qCE was 16.7% in both groups (see Appendix A).

Pair 2. During IE, more NW and OBE chose the small wide glass for big thirst and the wide glass for small thirst. However, there were no significant differences between the groups. During CE, significantly more OBE answered correctly (χ² (1, *n* = 87) = 5.14, *p* = 0.023). There were no significant differences between the groups in the qCE (see Appendix A).

Pair 3. During IE, more children in both groups chose the high narrow glass for big thirst and significantly more OBE chose the narrow glass for small thirst (χ² (1, *n* = 87) = 4.27, *p* = 0.039). The majority of the children intuitively decided correctly that the high narrow glass contained more water. During CE, significantly more OBE answered correctly (χ² (1, *n* = 87) = 4.36, *p* = 0.037). There was no significant difference between the groups in the qCE (see Appendix A).

#### Summary of Drinking Glasses

To summarize the results for the CE of the amounts in the three pairs of drinking glasses within the CE-score, significantly more OBE estimated the amount of water correctly (61%, 99 correct estimations out of 162) as compared to NW (43%, 35 correct estimations out of 81; χ² (1, *n* = 243) = 6.99, *p* = 0.008).

There was no significant difference in the reasons for choosing glasses for big vs. small thirst, with the most frequent reason being the difference in volume, followed by no specific reason, handiness, and habit (see Appendix A). When calculating the qCE-score across the three pairs of drinking glasses, there was no significant difference between OBE and NW (U(54,27) = 641.0, *p* = 0.378).

### 3.2. Assessment of the Amounts in Different Dishes

Pair 1. When the children had to choose the bowls according to big hunger, the result was ambiguous, with slightly more children of both groups choosing the big bowl. Significantly more NW chose the small bowl when feeling small hunger (χ² (1, *n* = 80) = 4.04, *p* = 0.044). For CE, the children could not distinguish the 20% more or less very well (see Table 4). When pouring the lentils (transfer task), the qCE was 16.7% for OBE and 12.5% for NW, which was not significant (see Appendix A).

Pair 2. For IE, there were no significant differences between the choices of OBE and NW. The majority of children chose the big plate for big hunger and the small plate for small hunger. During CE, more NW compared to OBE answered correctly. There was no significant difference in the qCE (see Appendix A).

Pair 3. For IE, there were no significant differences between the choices of OBE and NW. The majority of children chose the big plate for big hunger and the small plate for small hunger. During CE, significantly more NW answered correctly, compared to OBE (χ² (1, *n* = 80) = 6.67, *p* = 0.010). Moreover, NW performed better in the transfer task: there was a significant difference in the qCE between NW and OBE (11.4% vs. 23.1%; U(53,26) = 501.0, *p* = 0.031) (see Appendix A).

#### Summary of Dishes

To summarize the results for the CE of the amounts in the three pairs of dishes within the CE-score, significantly less OBE estimated the amount of lentils in the bowls and on the plates correctly (39%, 62 correct estimations out of 158) as compared to NW (56%, 45 correct estimations out of 80; χ² (1, *n* = 238) = 6.20, *p* = 0.013). 

There was no significant difference in the reasons for choosing bowls/plates for big vs. small hunger, with the most frequent reason being the difference in volume, followed by no specific reason, handiness, and habit (see Appendix A). When calculating the qCE-score across the three pairs of dishes, there was no significant difference between the OBE and NW (U(53,26) = 583.5, *p* = 0.268).

### 3.3. Additional Post-Hoc Analyses

From 24 h food-recall analyses, it is known that children of 10 years and older can give reliable information about the food intake [33]. To assess for possible age effects due to different developmental stages in youth, post-hoc analyses were conducted. The first analysis excluded the four youngest children aged between 9 and 10 years. The second analysis excluded the OBE children <10 and >14 years of age (the age range not represented by NW). In both analyses, the results were comparable with those of the complete analysis (see Appendix A).

## 4. Discussion

In this novel study, children and adolescents with obesity and normal weight were assessed in terms of portion size estimation in differently shaped dishware items for liquid and solid content. A special feature of the study is that the participants estimated the fluid and food quantities in the various dishware items both intuitively, via imaginary feelings of hunger and thirst, and cognitively. A clear finding of our experimental study is that the misjudgment of quantities by 20%, or even 30%, can happen very easily, depending on the serving container, and was shown in multiple populations including children and adolescents before [20,34]. It should be emphasized that both OBE and NW had varying degrees of uncertainty in estimating portion sizes. Therefore, the child’s and adolescent’s weight status appears to play a minor role in the perception of portion sizes and should be evaluated together with other factors. Furthermore, the role of dishware in estimating portion sizes has not yet been conclusively clarified and no recommendations can be derived. The results, however, provide an important indication that learning of appropriate portion sizes is also important for children and adolescents of normal weight, since larger portion sizes of energy-dense foods may result in increased daily energy intakes for both children and adults, suggesting their potential for increasing body weight over time [20,35].

OBE and NW estimated the amount of water in the three different drinking glass pairs relatively similarly when choosing glasses for big vs. small thirst (IE). More OBE estimated the quantities in the glasses correctly as compared to NW (CE-score: 61% vs. 43%) in the CE. OBE may have been exposed to (sweet) beverages in larger quantities than NW, but this hypothesis requires independent verification. Since excessive calories from beverages represent an easily adaptable driver of childhood obesity [36], it is particularly important to limit the amount of energy-dense beverages through portion sizes appropriate for children. However, for parents, it might be very difficult to choose the appropriate portion size for their children due to limited knowledge of appropriate portion sizes for children and themselves [37].

The situation was different for dishes (plates and bowls), where misjudgment played a particularly important role. A difficulty in estimating the amount of lentils could be that dried lentils are not ready to eat and children may only be familiar with the cooked form. In the intuitive approach, no differences between groups were detected when choosing a bowl/plate for big vs. small hunger, except for that more NW intuitively estimated the amounts of lentils in the small bowl at low hunger correctly compared to OBE. More NW estimated the amount of lentils in the bowls and on the plates correctly as compared to OBE during the CE (CE-score: 56% vs. 39%). This may be due to the fact that NW were more familiar with modest portion sizes on plates and in bowls, resulting in a more accurate estimation of quantity. Additionally, abnormalities of glucose and lipid metabolism might impede portion size estimation in obesity [38].

By using different strategies to shape and control children’s food intake, the parents might act as role models [39], which was not assessed within this study. Another important consideration is that some children might have lacked the necessary experience of handling food portions [40] due to a controlling feeding style from their parents, resulting in unclear decisions for an appropriate dishware item for small/big hunger/thirst, and non-targeted guessing of the amounts in/on the dishware. However, intuitive behavior is more likely to be habitual and was supported by answers such as “no specific reason” when asked why a particular plate/bowl was chosen. Furthermore, pathological eating behaviors as seen in eating disorders that usually develop during adolescence should be considered as a confounder in portion size estimation [41].

Age-appropriate dishware, including the use of smaller plates and plates with rims, was suggested for children [42], based on observations that adult-size plates encourage children to self-serve more of the foods they like, possibly by distorting their perception of physical size or by shifting normative beliefs about proper portion size [18]. On the other hand, larger plates can create more room for healthy food: using large-sized plates resulted in greater vegetable servings in adults [43]. Therefore, recommending the use of smaller plates might be premature, since a systematic review and meta-analysis demonstrated that there is no consistent effect of plate size on food intake in adults [19] and also not in children as discussed in a recent letter [23]. Additionally, the current investigation has shown that large as well as small plate sizes lead to misjudgments of portion sizes in children and adolescents. In children, the effect of serving food on smaller plates is not consistent [44]. In a study investigating the influence of plate size on the visual estimate of food portion size in forty-eight students, the size of the plate did not influence the estimate of food portions, but it did influence the classification of portion sizes [45]. The classification of portion sizes into small, medium and large portions might underlie the same instinctive pathways as the IE of our study and therefore, plate size does play a role when estimating the visual size of a portion. Especially amorphous foods, which take the shape of the container they are in, may be difficult to evaluate in terms of appropriate portion size [46]. Therefore, it is even more important for parents to support their children in serving proper portions since our study showed that especially OBE have difficulties in estimating the amount on differently shaped dishes.

## 5. Conclusions

This study demonstrated that estimating portion sizes in the context of classical dishware items such as glasses, bowls and dishes is challenging for children and adolescents independent of the weight status. Thus, the choice of dishware in a child/family setting may influence the perception of food amounts and therefore portion size served in children across the weight spectrum. Nevertheless, no clear recommendations can be given regarding dishware for home or take away meals due to a lack of data and the complexity of eating behavior and environmental factors. To date, it seems that experience with age-appropriate portion sizes and the energy density of foods may help best for decisions regardless of the size and shape of dishware and are extremely important in dietary training especially in OBE. Targeted research into these areas may provide powerful insight for primary treatment strategies and policy/environmental considerations for the management of childhood and adolescent obesity.

### Strength and Limits

In this study, parental feeding style, family environment or pre-test hunger ratings as confounding factors were not assessed. Furthermore, it was not evaluated whether the children serve themselves at mealtime, whether the food is served directly on plates/bowls at the table, or whether the children are allowed to take multiple servings or not; these factors may have had an influence on the results. Moreover, our experiment has a certain laboratory character, which did not include influential external food cues. On one hand, this is an advantage, since the elimination of external food cues allows assessing the isolated influence of dishware on portion size estimation. On the other hand, a laboratory does not represent a real-life condition. The study included a control group with normal-weight children and adolescents, which allowed us to analyze the selection of dishware in terms of weight status. Finally, the extent to which the findings generalize to children of other ages, cultures, and socioeconomic backgrounds requires further inquiry.

## Figures and Tables

**Figure 1 nutrients-13-02062-f001:**
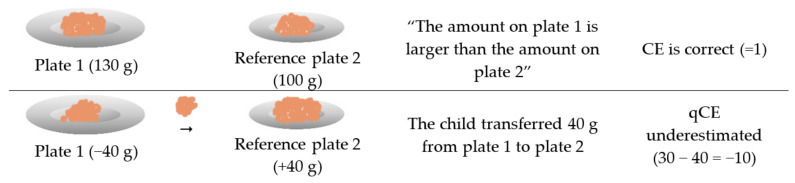
CE and qCE calculation shown as an example for one pair of plates. When calculating the scores, all three pairs of plates are included. Abbreviations: CE = Cognitive evaluation; qCE = Quantitative cognitive evaluation.

**Table 1 nutrients-13-02062-t001:** Characteristics of the study population.

	OBE	NW	*p*-Value
*n*	60	27	(χ^2^-test)
	(Mean ± SD)	(Mean ± SD)	(*t*-test)
Age (years)	13 ± 1.9	12.5 ± 0.9	0.093
Weight (kg)	84 ± 20.5	45.4 ± 8.2	<0.001
BMI (kg/m^2^)	31.2 ± 5.2	18.1 ± 1.6	<0.001
BMI-SDS	2.51 ± 0.57	−0.19 ± 0.55	<0.001

Results are presented as mean ± SD; Abbreviations: BMI = body mass index; BMI-SDS = body mass index-standard deviation score (z-score); NW = children and adolescents with normal weight; OBE = children and adolescents with overweight and obesity.

**Table 2 nutrients-13-02062-t002:** Characteristics of dishware items.

Pair	Dishware Item	Schematic Figure	Differencein Amount (%)	QuantityRatio
Glass pair 1	Wide parabolic glass, Q = 120vs.High parabolic glass, Q = 100	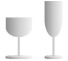	20	1.2:1
Glass pair 2	Small wide glass, Ø = 7.5, H = 9.0; Q = 250vs.Wide glass, Ø = 8.0, H = 13.5; Q = 250	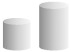	0	1:1
Glass pair 3	High narrow glass, Ø = 5.5, H = 15.5; Q = 300vs.Narrow glass, Ø = 5.5, H = 13.0; Q = 250	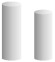	20	1.2:1
Dish pair 1	Big bowl, Ø = 15, Q = 180vs.Small bowl, Ø = 12, Q = 150	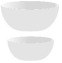	20	1.2:1
Dish pair 2	Big plate, Ø = 28, Q = 100vs.Small plate, Ø = 24, Q = 100	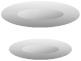	0	1:1
Dish pair 3	Big plate, Ø = 28, Q = 130vs.Small plate, Ø = 24, Q = 100	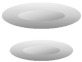	30	1.3:1

Ø = diameter in cm; H = height in cm; Q = quantity in mL (glasses) or g (dishes).

**Table 3 nutrients-13-02062-t003:** Results of the intuitive and cognitive evaluation for drinking glasses.

Pair	Glass (Content)	Intuitive Evaluation (IE)						Cognitive Evaluation (CE)				
		Big Thirst *n* (%)		*p*	Small Thirst *n* (%)		*p*	Correct *n* (%)		False *n* (%)		*p*
		NW	OBE		NW	OBE		NW	OBE	NW	OBE	
1	Wide parabolic (120 mL)	25 (92.6)	41 (68.3)	**0.014**	8 (29.6)	13 (22.0)	0.447	7 (25.9)	13 (22.0)	20 (74.1)	46 (78.0)	0.692
	High parabolic (100 mL)	2 (7.4)	19 (31.7)		19 (70.4)	46 (78.0)						
2	Small wide (250 mL)	16 (59.6)	38 (65.5)	0.577	6 (23.1)	20 (34.5)	0.296	12 (44.4)	40 (70.2)	15 (55.6)	17 (29.8)	**0.023**
	Wide (250 mL)	11 (40.7)	20 (34.5)		20 (76.9)	38 (65.5)						
3	High narrow (300 mL)	23 (88.5)	49 (86.0)	0.756	10 (38.5)	10 (17.5)	**0.039**	16 (59.3)	46 (80.7)	11 (40.7)	11 (19.3)	**0.037**
	Narrow (250 mL)	3 (11.5)	8 (14.0)		16 (61.5)	47 (82.5)						

Data presented as absolute incidence (percent); Significant *p*-values are shown in bold; Statistics: Chi-Square Tests; Abbreviations: *n* = absolute incidence; NW = children and adolescents with normal weight; OBE = children and adolescents with overweight and obesity; NW and OBE aged 9–17; *p* = *p*-value.

**Table 4 nutrients-13-02062-t004:** Results of the intuitive and cognitive evaluation for dishes.

Pair	Dishes (Content)	Intuitive Evaluation (IE)						Cognitive Evaluation (CE)				
		Big Hunger *n* (%)		*p*	Small Hunger *n* (%)		*p*	Correct *n* (%)		False *n* (%)		*p*
		NW	OBE		NW	OBE		NW	OBE	NW	OBE	
1	Big bowl (180 g)	13 (51.9)	32 (60.4)	0.297	2 (7.4)	14 (26.4)	**0.044**	14 (51.9)	24 (45.3)	13 (48.1)	29 (54.7)	0.578
	Small bowl (150 g)	14 (48.1)	21 (39.6)		25 (92.6)	39 (73.6)						
2	Big plate (100 g)	23 (85.2)	39 (73.6)	0.240	8 (29.6)	11 (20.8)	0.378	17 (63.0)	25 (47.2)	10 (37.0)	28 (52.8)	0.181
	Small plate (100 g)	4 (14.8)	14 (26.4)		19 (70.4)	42 (79.2)						
3	Big plate (130 g)	23 (85.2)	35 (66.0)	0.070	6 (22.2)	12 (23.5)	0.896	14 (53.8)	13 (24.5)	12 (46.2)	40 (75.5)	**0.010**
	Small plate (100 g)	4 (14.8)	18 (34.0)		21 (77.8)	39 (76.5)						

All data presented as absolute incidence (percent); Significant *p*-values are shown in bold; Statistics: Chi-Square Tests; Abbreviations: *n* = absolute incidence; NW = children and adolescents with normal weight; OBE = children and adolescents with overweight and obesity; NW and OBE aged 9–17; *p* = *p*-value.

## Data Availability

Not applicable.

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
