# Peer review of "The Role of Dishware Size in the Perception of Portion Size in Children and Adolescents with Obesity"

_nutrients, 2021, doi:10.3390/nu13062062_

Round 1

Reviewer 1 Report

This manuscript investigated how children and adolescents with normal weight or obesity perceived portion sizes in differently shaped dishware items.  The study is interesting although lacks external validity and maybe limited in scope.  I have the following specific comments.

The age range of the children was between 9-17 years and there are major changes in physical, cognitive and psychological factors within this age range.  Were differences in development accounted for?   How did age influence the understanding of concepts such as little or great hunger?  Had the older children more opportunities to learn about portion sizes?

The introduction may benefit from a discussion of expected satiety.

Had the children had experience eating lentils?  Could they make an accurate estimate regarding the amount of dried lentils (lentils are not consumed in this manner although I appreciate why the authors used this approach) required to satiate a greater hunger?   

References – references often use ‘und’ rather than ‘and’.

Abstract

Line 18 – I would recommend rephrasing the sentence so that hunger or thirst are not described as great or little (this is throughout the manuscript)

Introduction

line 13-14 – Eating frequency may also be a key contributor to weight management.

Line 21- 23 – suggest changing this to state that caregivers serving sizes are strongly associated with the amount eaten and not the determinant of the amount eaten.

Line 61 – suggest using “hypothesize” rather than expect.

Methods

The different dishware were presented to participants in a standardized manner – was the order in which the dishware was presented randomized or were all dishware presented in the same order?

Line 127 - Define the term ‘handiness’

Were the participants informed about the purpose of the study?

Line 148 – please provide a version number and reference for G-Power.

Discussion

In the introduction, it is stated that the amount that a caregiver serves is an important determinant of food intake.  However, in the discussion the authors discuss the use of age-appropriate plates.  Does the size of the dishware matter to children (or their perception of dishware size) if they do not serve themselves?

Author Response

REVIEWER 1:

This manuscript investigated how children and adolescents with normal weight or obesity perceived portion sizes in differently shaped dishware items.  The study is interesting although lacks external validity and maybe limited in scope.  I have the following specific comments.

The age range of the children was between 9-17 years and there are major changes in physical, cognitive and psychological factors within this age range.  Were differences in development accounted for?   How did age influence the understanding of concepts such as little or great hunger?  Had the older children more opportunities to learn about portion sizes?

Thank for this important point. We included post-hoc analysis in order to deal with this issue. The data are also presented in the supplements: “From 24 h food-recall analyses it is known that children of 10 years and older can give reliable information about the food intake (Biró et al. 2002). To assess for possible age effects due to different developmental stages in youth, post-hoc analyses were conducted. The first analysis excluded the four young-est children aged between 9 and 10 years. The second analysis excluded the OBE children <10 and >14 years of age (the age range not represented by NW). In both analyses, the results were comparable with those of the complete analysis (see supplementary resources 3 – 7).”

The introduction may benefit from a discussion of expected satiety.

Thank you for this suggestion. We have included a brief sentence about the role of expected satiety in the introduction: “Moreover, the children’s ability to estimate the satiating properties of foods (Hardman et al. 2011) might also represent an important factor when choosing dishware as well as portion sizes.”

Had the children had experience eating lentils?  Could they make an accurate estimate regarding the amount of dried lentils (lentils are not consumed in this manner although I appreciate why the authors used this approach) required to satiate a greater hunger?   

Thank you very much for this important aspect. We included this topic in the discussion: “A difficulty in estimating the amount of lentils could be that dried lentils are not ready to eat and children may only be familiar with the cooked form.”

References – references often use ‘und’ rather than ‘and’.

This has been corrected – thank you.

Abstract

  • Line 18 – I would recommend rephrasing the sentence so that hunger or thirst are not described as great or little (this is throughout the manuscript)

Thank you very much for this suggestion. After getting advice from a native speaker, we changed great/little to big/small.

  • line 13-14 – Eating frequency may also be a key contributor to weight management.

We totally agree that eating frequency is an important aspect with regard to weight management. Therefore, we have included it in our introduction: “From a nutritional point of view, portion size and energy density (Rolls 2017; Mack et al. 2020) of foods as well as eating frequency (Kant 2014) are key factors of weight management (Pourshahidi et al. 2014).”

  • Line 21- 23 – suggest changing this to state that caregivers serving sizes are strongly associated with the amount eaten and not the determinant of the amount eaten.

Thank you for this suggestion. We have changed the sentence: “In the home setting, caregiver serving sizes are strongly associated with the amounts children consume (Johnson et al. 2014).

  • Line 61 – suggest using “hypothesize” rather than expect.

“We hypothesize that there will be differences between NW and OBE in the estimation of…”

Methods

  • The different dishware were presented to participants in a standardized manner – was the order in which the dishware was presented randomized or were all dishware presented in the same order?

The questions and the process were always the same. It was started with the plates, next came the glasses. Within these groups, the children were allowed to choose which pair of plates or glasses they wanted to start and precede with. I.e. indirectly it was randomized because the children could decide the order freely. The sequence (first plates, then glasses) and the order of the questions remained the same. This information is now also provided in the methods section.

  • Line 127 - Define the term ‘handiness’

Thank you for this aspect. We agree that the term “handiness” is difficult to understand in this context. That is why we have changed it to “easy handling”.

  • Were the participants informed about the purpose of the study?

Yes, the participants were informed about the study’s purpose: “The procedure started in the afternoon around 15:00, approximately 2 hours after lunch and took 30-45 minutes. The participants were informed about the purpose of the study.”

  • Line 148 – please provide a version number and reference for G-Power.

Done.

Discussion

  • In the introduction, it is stated that the amount that a caregiver serves is an important determinant of food intake.  However, in the discussion the authors discuss the use of age-appropriate plates.  Does the size of the dishware matter to children (or their perception of dishware size) if they do not serve themselves?

Thank you for this important point. We have stated in the introduction that self-serving is an important determinant of portion size. If children do not serve themselves, they have to learn to focus on the food amount rather than on the plate size. However, this is very difficult, since we tend to use dishware sizes as norms for portion sizes. We have added the aspect that for parents, it might be very difficult to chose appropriate portion sizes for their children – therefore the use of age-appropriate plates might be a support for parents. “However, for parents it might be very difficult to choose appropriate portion size for their children due to limited knowledge of appropriate portion sizes for children and themselves (Kairey et al. 2018).”

Thank you for your time and efforts which have helped to improve the quality of the manuscript.

Reviewer 2 Report

The Authors decided to evaluate children’s perception of dishware in regards to the portion size. It is an innovative and interesting approach to the problem of childhood overweight and obesity. The presence of the normal-weight control group in the study should be highlighted, as the results of two-armed studies are generally regarded as evidence of relatively higher quality.

Detailed comments from the Reviewer are provided below.                     

Abstract:

- adding the range of calendar age of the examined children to the “Methods” part of the abstract would be helpful.

Methods:

- Table 1. – it would be beneficial to provide information regarding the percentage of children with overweight and with obesity in examined group, just as additional information for the reader;

- line 122 – there is a statement “For assessing IE, we asked all children…”; generally, scientific texts should be written in an impersonal form (i.e. children were asked), therefore changing this and similar statements to impersonal form will help to improve the overall quality of the text;

- lines 137-140 – the Authors state, that the process used to verify the participants’ cognitive ability to distinguish between size dimensions was based on the ability to differentiate between three different everyday objects; in the Reviewer’s opinion, it will be useful to indicate what objects were used for this task;

- the Authors include children and adolescent aged 7-17, which isa group quite diverse in terms of development, therefore also cognitive abilities, and thus may differ in regards to the ability to assess the food/ drink portions; in the Reviewer’s opinion it will be helpful to analyse the results separated into different age groups and maybe present them in the supplement; not necessarily one-year ones, but perhaps two/ three categories (for example pre-, peri- and postpubertal or pre-and postpubertal categories).

Discussion:

- the Authors refer to the participants with excess body weight as having obesity (line 248) or overweight (257) using these terms interchangeably; as the examined group consisted of both categories, in the Reviewer’s opinion, it will be better to use one term, including both of those, such as excess body weight or previously used abbreviation;

- line 325 – there is a spelling error - “bowels” should be “bowls”.

Author Response

REVIEWER 2:

The Authors decided to evaluate children’s perception of dishware in regards to the portion size. It is an innovative and interesting approach to the problem of childhood overweight and obesity. The presence of the normal-weight control group in the study should be highlighted, as the results of two-armed studies are generally regarded as evidence of relatively higher quality.

Thank you very much for this suggestion. We have added the aspect of the presence of a control group in the introduction “Twenty-seven healthy children with normal weight (NW) with a BMI between the 10th and ≤85th BMI percentile and aged between 11 and 14 years (56% male) living in the area around the University Hospital Tübingen were included as a control group.”, and in the discussion “The study included a control group with normal weight children and adolescents, which allowed to analyze the selection of dishware in terms of weight status.”

Detailed comments from the Reviewer are provided below.                    

Abstract:

  • adding the range of calendar age of the examined children to the “Methods” part of the abstract would be helpful.

Thank you, this is a very helpful suggestion: “The study included 60 children and adolescents with overweight and obesity (OBE) and 27 children and adolescents with normal weight (NW) aged 9 to 17 years.”

Methods:

  • Table 1. – it would be beneficial to provide information regarding the percentage of children with overweight and with obesity in examined group, just as additional information for the reader;

Thank you for this suggestion; as only 5 children were classified as overweight, we decided to use the category “obesity” for the OBE group: “Five children were considered overweight, which represent <10% of the OBE group. Therefore only the category obesity is used in the following text.”

  • line 122 – there is a statement “For assessing IE, we asked all children…”; generally, scientific texts should be written in an impersonal form (i.e. children were asked), therefore changing this and similar statements to impersonal form will help to improve the overall quality of the text;

Thank you for this helpful comment. We have changed our text into impersonal form.

  • lines 137-140 – the Authors state, that the process used to verify the participants’ cognitive ability to distinguish between size dimensions was based on the ability to differentiate between three different everyday objects; in the Reviewer’s opinion, it will be useful to indicate what objects were used for this task;

Thank you. This information is now provided in the manuscript: “As a validation step, each child’s cognitive ability to distinguish between size dimensions of three everyday objects was tested (mobile phone (9 cm), book (24 cm), bottle (34 cm)).”

  • the Authors include children and adolescent aged 7-17, which is a group quite diverse in terms of development, therefore also cognitive abilities, and thus may differ in regards to the ability to assess the food/ drink portions; in the Reviewer’s opinion it will be helpful to analyse the results separated into different age groups and maybe present them in the supplement; not necessarily one-year ones, but perhaps two/ three categories (for example pre-, peri- and postpubertal or pre-and postpubertal categories).

Thank for this important comment. We agree and included post-hoc analysis in order to deal with this issue. The data are also presented in the supplements: “From 24 h food-recall analyses it is known that children of 10 years and older can give reliable information about the food intake (Biró et al. 2002). To assess for possible age effects due to different developmental stages in youth, post-hoc analyses were conducted. The first analysis excluded the four young-est children aged between 9 and 10 years. The second analysis excluded the OBE children <10 and >14 years of age (the age range not represented by NW). In both analyses, the results were comparable with those of the complete analysis (see supplementary resources 3 – 7).”

We analyzed the summarized CE score for the different age categories. For the three pairs of dishes, the results remained the same (NW performed better) after excluding the four youngest children aged between 9 and 10 and when separately analyzing the group of 10- to 14-year olds (see line 260-263).

For the glasses, when excluding the four youngest children, results remained the same (OBE performed better). When analyzing the 10-14 year old children, there were no significant differences between NW and OBE, but the clear statistical trend (p = 0.064) towards OBE performing better remained (see line 221 – 225). However, we also have to bear in mind that the sample size was reduced for this sub-analysis, decreasing the statistical power.

We believe that these post-hoc analyses improve the manuscript. We did not perform further subgroup comparisons because the sample size would have decreased rapidly (especially for the NW) which would have further reduced the statistical power.

Discussion:

  • the Authors refer to the participants with excess body weight as having obesity (line 248) or overweight (257) using these terms interchangeably; as the examined group consisted of both categories, in the Reviewer’s opinion, it will be better to use one term, including both of those, such as excess body weight or previously used abbreviation;

Thank you for this suggestion. We have corrected the terms throughout the text.

  • line 325 – there is a spelling error - “bowels” should be “bowls”.

Done.

Thank you for your time and efforts which have helped to improve the quality of the manuscript.

Round 2

Reviewer 1 Report

Thank you for taking the time to carefully address my comments.  This is an interesting paper.